# Tree Growth Conditions Are Demanded When Optimal, Are Unwanted When Limited, but When Are They Suboptimal?

**DOI:** 10.3390/plants10091943

**Published:** 2021-09-18

**Authors:** Claudia Cocozza, Maria Laura Traversi, Alessio Giovannelli

**Affiliations:** 1Department of Agriculture, Food, Environment and Forestry, Via San Bonaventura 13, I-50145 Florence, Italy; claudia.cocozza@unifi.it; 2CNR—Institute of Research on Terrestrial Ecosystems, Via Madonna del Piano 10, I-50019 Sesto Fiorentino, Italy; marialaura.traversi@cnr.it

**Keywords:** acclimation, phenotypic plasticity, climate change, ecosystem services, epigenetic mechanisms

## Abstract

The recent climate projections predict that the intensity and frequency of extreme events will increase as a result of overall increasing mean temperature and reduced precipitations in the temperate regions of the Northern Hemisphere. How these changes will influence the harshness of the environment and the performances of trees growing under natural conditions remains an open question. In this commentary article, we would like to look at the concept of suboptimal growth conditions, widening its application from the traditional in vitro manipulation to trees growing in open air, addressing the main limitations and strengths of the upscaling results from cell to tree. We believe that the traditional single dose–effect approach is not suitable to explain the complex interactions between genotype and environment, occurring in open field or forest stands, where the intensity and frequency of the events are uncontrolled and unpredictable. As forests provide a wide range of ecosystem services, new parameters should be considered in the definition of the response thresholds in addition to growth. Thus, within this Special Issue, we stimulate the discussion over the development of new approaches and technologies that are able to define suitable threshold responses of trees under suboptimal natural conditions, with the aim to furnish new insights on the acclimation and adaptation processes in woody species under global change.

## 1. Toward a Wider Definition of ‘Suboptimal Growth Conditions’

The term ‘tree suboptimal growth conditions’ is often used in biology and ecophysiology to identify the environmental conditions surrounding the plants, able to affect their growth or yield and to induce response mechanisms. The modulation of the plant/organ/cell response to suboptimal conditions depends on the nonlinear interaction effects of the following two components: the genetic fitness (traits from genetic evolution) that determine the ‘stress sensitivity’, defined as the level threshold of the abiotic factor, inducing a physiological response of the plant; the environmental conditions that determine the ‘stress intensity’, defined as the level of harshness of the surrounding conditions of the plant. Each species/plant/organ has its own response threshold to the environmental factors, and it can change over the time (ontogenesis) or as a function of position within the trees (topophysis). Ideally, each threshold response defines tree growth conditions as follows: optimal (genetic fitness prevails over the environment); suboptimal (genetic fitness and environment interact); or survival (environment prevails over the genetic fitness) (Figure 1a). In agriculture practices, the suboptimal conditions are sometimes induced to increase the yield (Figure 1b). Irrigation deficit, chilling, pruning, and shading are some of the common practices used to improve crop yield under suboptimal conditions. Thus, the achievement of suboptimal conditions cannot always be a detrimental condition for our needs. On the same side, the breeding programs are moving towards the selection of genotypes with the lowest stress sensitivity, which may assure high resilience in a dynamic environment, maintaining a high yield and reducing the use of resources [1]. The concept of suboptimal conditions assumes contrasting meanings if it is applied to short-term manipulative experiments under controlled environments, or to the long-term monitoring of natural or managed forest stands subjected to unpredictable climate events and competition. Under controlled environments, the effect of a single parameter variation can be tested in a defined experimental design, and the dose–effect response of cells, organs, and plants to an exogenous application (macro and micronutrients, growth regulators, organic matter, carbon sources, etc.), or to the modulation of environmental parameters (temperature, light, water, soil pH, etc.), can be evaluated. The use of growth chambers or ‘in vitro techniques’ were often proposed as suitable tools to assure replicability and reproducibility of the results [2,3], even if the upscale of these results to open field remains complex nowadays. The confounding interactions between environmental factors, related to their variability and intensity, determine unpredictable shifts in the tree response, which can lead to tree death [4]. In a recent review, Dubois and Inzé [2] highlighted the needs to bridge the gap between short-term in vitro results with long-term soil-based experiments. In natural or managed forests, the “optimal” is pragmatically inferred from the environmental conditions in which the species usually occur [5], and “suboptimal” to environmental phenomena that are able to reduce trees performances or to negatively affect ecosystem services. Thus, in forest stands, it could be more important to understand how species differ in their capacity, to follow shifts in their optimal conditions (i.e., conditions of maximal growth and minimal stress) or to persist under suboptimal conditions (i.e., conditions of reduced growth, due to increased stress strength) [6], rather than to focus on the dose–growth response approach. This raises the question of what the relevant physiological traits are able to discriminate the contrasting behaviors (following vs. persisting strategies) under suboptimal conditions [5].

## 2. How to Study the Effects of Suboptimal Conditions in Trees under Natural Conditions

To decipher the tree responses to withstand environmental constraints, small- and large-scale, short- and long-term, in vivo and in vitro experiments were performed in the past [2]. In these ways, the suboptimal conditions of growth were established along theresources gradient to test, for instance, the implications of shifting temperatures and soil moisture on tree functions. Experimental setups vary from ‘test’ plant species to multiple ‘test’ species. The context in which any plant growth experiment is conducted is almost as important as the outcome. The choice of experimental conditions allows the responses of the growth/ecosystem services to be highlighted under suboptimal conditions, as well as the above and below environment thresholds that induce acclimation to be identified [7]. Experiments allow the performance of trees to be determined from visual assessments [8] to quantitative measures of plant growth [9,10,11], as well as at physiological and transcriptome levels. High-throughput experimental systems, e.g., greenhouse and growth chamber conditions, are used to punctually, continuously, and fastly monitor whole-plant responses to set-up growth conditions. Whereas experiments under open field conditions might take longer, by assuming the delayed reaction of plant responses, due to uncontrolled environmental conditions. Thus, latitudinal and altitudinal transects in forest stands, as well as common gardens, are suitable approaches to study acclimation strategies through observational or manipulative experiments [12]. In these cases, remote sensing, digital aerial and terrestrial photogrammetry, as well as aerial or terrestrial laser scanning, can also be used for linear or volumetric measurements of the trees, and to assess forest health by the use of the vegetation index [13,14]. In the last years, several studies have showed that the epigenetic mechanisms (mechanisms affecting gene expression without DNA mutation, and that can sometimes be hereditable) have a crucial role in the acclimation, memory, priming, and adaptive plasticity in trees [15]. The epigenetic changes can be investigated nowadays by traditional methods, such as DNA methylation or advances in epigenomic editing approaches [16].

## 3. How Does Global Change Affect Suboptimal Conditions at Large Scale?

Recent climate projections predict that the global mean temperature will rise from 1.7 to 4.8 °C by 2100 [17], with warming being more pronounced at high latitudes of the Northern Hemisphere [18,19]. It is postulated that the increase in air temperature will affect the water cycle at the local scale, and an increase in precipitation is expected in cold temperate and boreal regions, because of the higher water vapor concentration in the troposphere. On the contrary, a decrease in precipitation is expected in arid and semi-arid regions, with an increase in the likelihood of extreme events [20]. How these changes will influence the harshness of environmental conditions in these regions remains an open question [21]. In this scenario, recent evidence highlighted that woody species located at high latitudes and alpine areas could benefit from global warming because of a growing season lengthening, while growth in the southern regions might be negatively affected by warming and drought, inducing some species to shift towards suboptimal conditions [22]. Thus, a crucial question arises of which species will be more prone to persist in harsh environments up to survival thresholds, and which one will follow the optimal growth conditions [5].

## 4. Might Suboptimal Conditions Drive Tree Acclimation and Adaptation?

Trees control growth and functionality daily, by regulating photosynthesis to gain carbon and respiration to loss carbon. These processes are mainly affected by ambient temperatures and water availability. Physiology across tree species underlies the differences in functional traits and growth responses, due to different mechanisms of acclimation and adaptation. Therefore, trees adopt acclimation mechanisms to maintain optimal performance under suboptimal growth conditions that, according to long-term changes, become adaptation mechanisms to withstand extreme and persistent environmental events [23]. For these reasons, tree adaptation has drawn ecological zones where ecological classifications established the most-favorable sites for tree species, and those not suitable as the dryer and the wetter sites [24]. Moreover, adaptation processes to temporal changes in water availability determine ecological niches of species, where differences in the regeneration niches reflect different physiological adjustments caused by the parental supply of resources [25]. 

## 5. General Remarks

Suboptimal conditions for tree growth are essential to understand the dynamics of forest evolution from optimal to stressful environments. These findings highlight the importance of tree resilience to site conditions, by providing an indication of the competition in suboptimal conditions. Recent findings showed that epigenetic responses can be considered as one of the main drivers of the acclimation and adaptive plasticity in trees. Based on these considerations, some of the following questions arise: What are the environmental thresholds able to induce stress memory and acclimation? What are the intensities/durations/frequencies of environmental factors able to induce epigenetic responses? Could the crops growing under controlled suboptimal conditions (for example, crop submitted to irrigation deficit) be affected by epigenetics effects? The identification of suboptimal thresholds of environmental resources might permit the identification of “tree ecotypes” under different growth conditions, by suggesting an opportunity for future efforts aimed at modulating the forest sustainable management. In this frame, dendroclimatic approaches can be used to assess phenotypic plasticity in forest stands. Furthermore, new efforts should be carried out to select relevant functional traits that are able to define the ecosystem service values of forests growing under suboptimal growth conditions, to furnish new information to drive the next forest management programs. For this reason, multidisciplinary studies should be set, involving different areas of science, from the traditional tree genetics, physiology, and dendroclimatology to the emergent research fields, such as remote sensing and metabolomics.

## Figures and Tables

**Figure 1 plants-10-01943-f001:**
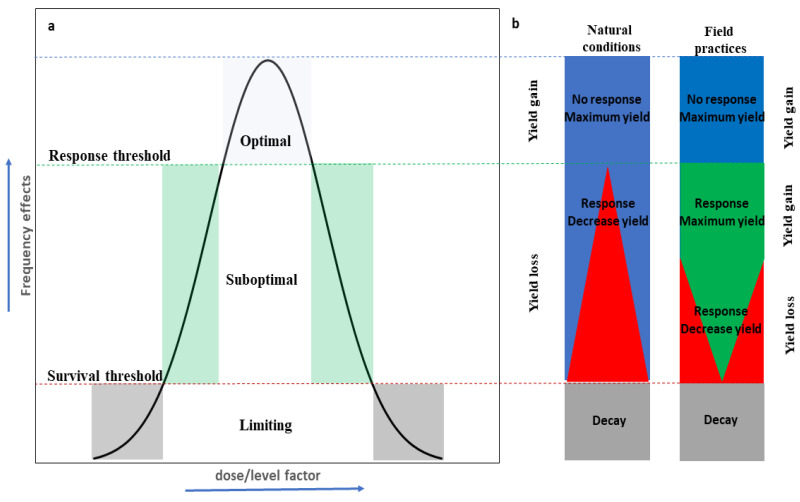
Theoretical representations of suboptimal growth conditions referred to a hypothetical linear change in the abiotic factor level. (**a**) Optimal growth conditions (blue area) represent the range of dose/level factor able to maximize the yield without inducing physiological responses (response threshold) of tree/organ/cell. Under optimal conditions, the genotype effect prevails over the environment and trees express their maximum genetic potential. Suboptimal growth conditions (green area) represent the changing environment in which range of dose/level factors can induce physiological response in trees/organs/cells (a part of energy is consumed to face with the changing environment). Under suboptimal conditions, the interaction between environment and genotype drives the response, and trees express their phenotypic plasticity. Under limiting conditions (grey area), trees/organs/cells are unable to face with changing environments as dose/level factors are too low or high to allow acclimation and tree can undergo decay. (**b**) Hypothetical effect of optimal, suboptimal, and limiting growth conditions on the physiological response of trees determining yield loss and yield gain. Under optimal conditions, the trees express their genetic potential, and maximum yield is reached (blue area). Under suboptimal conditions (blue/red area), the increasing/decreasing in dose/level factor induces physiological responses that increase as a function of severity of the environmental changes and produce yield loss. Sometimes suboptimal conditions are artificially applied in managed field (yield gain), and physiological response (green area) is used to improve yield (deficit irrigation, pruning, girdling, shading treatments). In both cases, the trees can rapidly undergo decay below the survival threshold. The linear increasing in the yield loss (red area) is not realistic, but it is an oversimplification for graphic representation.

## Data Availability

Not applicable.

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
