# Peer review of "Tree Growth Conditions Are Demanded When Optimal, Are Unwanted When Limited, but When Are They Suboptimal?"

_plants, 2021, doi:10.3390/plants10091943_

Round 1

Reviewer 1 Report

The reviewed work contains very important information on the growth conditions of prevailing trees in various ranges of ecological tolerance. At work i.a. 'suboptimal growth conditions' were defined and the possibilities of testing these conditions in natural phytocoenoses, the impact of global changes on suboptimal growth conditions were described. The knowledge review can be used, inter alia, in dendroclimatic research.

The work may be published unmodified in the Plants journal.

Author Response

We are very gratefull to the reviewer for the positive comments. We agree with the involvement of dendroclimatic researches in the study of suboptimal conditions in forest trees. For this we added the following sentence (line 169-170 new version) ‘In this frame, dendroclimatic approaches can be used to assess phenotypic plasticity in forest stands.’

Reviewer 2 Report

Dear authors, I found your manuscript is interesting and informative for the commentary for the Special Issue. The manuscript is well written, clear, and brings to the reader’s attention important topical questions. My only remark is, could you name which areas of science can potentially search for answers to the raised questions (ecophysiology, plant physiology, etc.)? In such a way, present to the reader the scale of interdisciplinary studies that will be collected in the Issue.

Author Response

We are very gratefull to the reviewer for the positive comment and the important suggestions about the putative area of science of interest and the multidisciplinary approach. Following the comment of the reviewer we added the following sentence: (line 173-176 new version). For this reason, multidisciplinary studies should be set involving different area of science from the traditional tree genetics, physiology and dendroclimatology up to the emergent research fields like remote sensing and metabolomics.’.

Reviewer 3 Report

I recommend publishing the manuscript ID: plants-1379417 entitled "Tree growth conditions are demanded when optimal, are unwanted when limited, but when they are suboptimal?" in Journal "Plants" ISSN: 2223-7747 as a supporting commentary of the authors on their special issue. The authors provide a more detailed overview of the chosen topic with the support of literary sources.

My suggestion for improvement: If the authors consider it appropriate, I recommend including methods of remote sensing in the investigation of the health and growth of plant species. Remote sensing, Digital aerial and terrestrial photogrammetry and aerial or terrestrial laser scanning can also be used for linear or volumetric measurements of the trees etc. There is a lot of research on this topic.

I wish the authors a lot of inspirative articles in their special issue and success in their own research. 

Author Response

We are very gratefull to the reviewer for the positive comments and we appreciated the suggestion about the use of remote sensing as suitable tool to study suboptimal condition in forest. So, following the suggestion of the reviewer we added this sentence and two references: (line 117-120 new version of the ms) ‘In these cases, remote sensing, digital aerial and terrestrial photogrammetry as well as aerial or terrestrial laser scanning can also be used for linear or volumetric measurements of the trees and for assessing forest health by use of vegetation index [13, 14].